# Time-Dependent Efficacy of Checkpoint Inhibitor Nivolumab: Results from a Pilot Study in Patients with Metastatic Non-Small-Cell Lung Cancer

**DOI:** 10.3390/cancers14040896

**Published:** 2022-02-11

**Authors:** Abdoulaye Karaboué, Thierry Collon, Ida Pavese, Viviane Bodiguel, Joel Cucherousset, Elda Zakine, Pasquale F. Innominato, Mohamed Bouchahda, René Adam, Francis Lévi

**Affiliations:** 1Medical Oncology Unit, GHT Paris Grand Nord-Est, Le Raincy-Montfermeil, 93770 Montfermeil, France; thierry.collon@ght-gpne.fr (T.C.); ida.pavese@ght-gpne.fr (I.P.); 2UPR “Chronotherapy, Cancer and Transplantation”, Medical School, Paris-Saclay University, 94800 Villejuif, France; Pasquale.Innominato@wales.nhs.uk (P.F.I.); mohamed.bouchahda@avec.fr (M.B.); rene.adam@aphp.fr (R.A.); 3Pathology Unit, GHT Paris Grand Nord-Est, Le Raincy-Montfermeil, 93370 Montfermeil, France; viviane.bodiguel@ght-gpne.fr (V.B.); joel.cucherousset@ght-gpne.fr (J.C.); elda.zakine@ght-gpne.fr (E.Z.); 4North Wales Cancer Centre, Ysbyty Gwynedd, Betsi Cadwaladr University Health Board, Bangor LL57 2PW, UK; 5Cancer Chronotherapy Team, Cancer Research Centre, Division of Biomedical Sciences, Warwick Medical School, Coventry CV4 7AL, UK; 6Medical Oncology Department, Paul Brousse Hospital, 94800 Villejuif, France; 7Medical Oncology Unit, Clinique Saint Jean L’Ermitage, 77000 Melun, France; 8Medical Oncology Unit, Clinique du Mousseau, 91000 Evry, France; 9Centre Hépato Biliaire, AP-HP, Hôpital Paul Brousse (APHP), 94800 Villejuif, France

**Keywords:** non-small-cell lung cancer, checkpoint inhibitors, nivolumab, circadian timing, overall survival

## Abstract

**Simple Summary:**

Initial clinical observations revealed strikingly longer follow-up for metastatic non-small-cell lung cancer (NSCLC) patients receiving nivolumab infusions predominantly in the morning as compared to those treated in the afternoon. Prior experimental and human studies have demonstrated the temporal distributions of immune cells’ proliferation, trafficking, and antigen recognition and destruction over the 24 h. Here, we hypothesized that circadian timing could play an important role in nivolumab’s efficacy, as previously shown for the toxicity and/or efficacy of chronomodulated chemotherapy in colorectal and lung cancer patients. Following project validation by an internal scientific review board, the dosing times of each of the 1818 nivolumab infusions given to 95 consecutive patients as a standard treatment for metastatic NSCLC were retrieved from the day-hospital records. Adverse events and radiologically documented tumor responses were retrieved and reviewed from patients’ clinical charts. Patients were allocated to ‘morning’ (N = 48 patients) or ‘afternoon’ (N = 47 patients) groups, according to whether they had received the majority of nivolumab infusions before or after 12:54, i.e., the median time of all infusions, respectively. ‘Morning’ nivolumab dosing nearly quadrupled median progression-free and overall survival as compared to ‘afternoon’ dosing. ‘Morning’ nivolumab was significantly more effective irrespective of age, sex, performance status, prior treatments, tumor histology, or PD-L1 expression. In contrast, nivolumab primary resistance was most often observed following ‘afternoon’ dosing. Randomized trials are warranted both to further identify the optimal timing of checkpoint inhibitors in individual cancer patients, and to determine the main mechanisms that precisely drive immunotherapy efficacy and resistance along the circadian timescale.

**Abstract:**

Hypothesis: Prior experimental and human studies have demonstrated the circadian organization of immune cells’ proliferation, trafficking, and antigen recognition and destruction. Nivolumab targets T(CD8) cells, the functions, and trafficking of which are regulated by circadian clocks, hence suggesting possible daily changes in nivolumab’s efficacy. Worse progression-free survival (PFS), and overall survival (OS) were reported for malignant melanoma patients receiving more than 20% of their immune checkpoint inhibitor infusions after 16:30 as compared to earlier in the day. Methods: Consecutive metastatic non-small-cell cancer (NSCLC) patients received nivolumab (240 mg iv q 2 weeks) at a daily time that was ‘randomly’ allocated for each course on a logistical basis by the day-hospital coordinators. The median time of all nivolumab administrations was computed for each patient. The study population was split into two timing groups based upon the median value of the median treatment times of all patients. CTCAE-toxicity rates, iRECIST-tumor responses, PFS and OS were computed according to nivolumab timing. PFS and OS curves were compared and hazard ratios (HR) were computed for all major categories of characteristics. Multivariable and sensitivity analyses were also performed. Results: The study accrued 95 stage-IV NSCLC patients (PS 0–1, 96%), aged 41–83 years. The majority of nivolumab administrations occurred between 9:27 and 12:54 for 48 patients (‘morning’ group) and between 12:55 and 17:14 for the other 47 (‘afternoon’ group). Median PFS (95% CL) was 11.3 months (5.5–17.1) for the ‘morning’ group and 3.1 months (1.5–4.6) for the ‘afternoon’ one (*p* < 0.001). Median OS was 34.2 months (15.1–53.3) and 9.6 months (4.9–14.4) for the ‘morning’ group and the ‘afternoon’ one, respectively (*p* < 0.001). Multivariable analyses identified ‘morning’ timing as a significant predictor of longer PFS and OS, with respective HR values of 0.26 (0.11–0.58) and 0.17 (0.08–0.37). The timing effect was consistent across all patient subgroups tested. Conclusions: Nivolumab was nearly four times as effective following ‘morning’ as compared to ‘afternoon’ dosing in this cohort of NSCLC patients. Prospective timing-studies are needed to minimize the risk of resistance and to maximize the benefits from immune checkpoint inhibitors.

## 1. Introduction

The recent advent of immunotherapy represents a major milestone for the improvement of the outcomes of patients with advanced or metastatic non-small-cell lung cancer (NSCLC) [1,2,3]. Thus, several randomized controlled trials have demonstrated large and statistically significant prolongation of both progression-free survival (PFS) and overall survival (OS) with antibodies directed against programmed death 1 (PD-1) or programmed death-ligand 1 (PD-L1) as compared to cytotoxic chemotherapy in lung cancer patients [1,4,5,6]. As a result, nivolumab has become the first immunotherapeutic agent approved as the standard of care for patients with metastatic or advanced NSCLC who progressed during or after treatment with platinum-based chemotherapy [7,8].

Nivolumab is a fully human immunoglobulin G4 monoclonal antibody that binds to programmed death-1 (PD-1) receptors and blocks their interaction with programmed death-ligand 1 (PD-L1) and programmed death-ligand 2 (PD-L2), thereby preventing activation of the PD-1 pathway [9]. PD-1 is highly expressed on the surface of tumor-specific T (CD8) lymphocytes, as well as on other activated T, natural killer and B lymphocytes, macrophages, dendritic cells and monocytes [9]. Nivolumab mainly enhances the activity of T cells against tumor cells. It has long been known that both the functional activity and trafficking of immune cells are regulated over the 24 h by molecular circadian clocks expressed by these cells [10,11,12,13,14,15,16,17]. This is notably the case for T(CD8) lymphocytes, which represent the main target cells of nivolumab [18]. Based on several experimental and human studies, our team earlier hypothesized a circadian coordination of various immune functions, notably with the highest proliferation of immune cells in the early afternoon, increased blood circulation in the late evening for B-cells and near the middle of the night for T-cells, and antigen recognition and destruction processes predominant in the early morning hours [19]. These and the large body of more recent data [10,11,12,13,14,15,16,17] suggest a possible impact of dosing time of nivolumab on its efficacy. Several studies have mapped circadian rhythms in NSCLC patients, including those in circulating lymphocyte subsets, and others have shown the relevance of circadian timing for chemotherapy tolerability [20,21,22,23,24,25]. Finally, recent data highlighted the clinical relevance of the timing of administration of immune checkpoint inhibitors (ipilimumab, nivolumab, or pembrolizumab, or a combination of these) in metastatic melanoma patients [26].

The present study was based on an initial clinical observation of a strikingly longer follow up of metastatic NSCLC on second-line nivolumab treatment administered in the morning as compared to the afternoon. We then hypothesized that circadian timing could play an important role in nivolumab’s efficacy, based on our prior trials of chronomodulated chemotherapy [27,28,29]. Strikingly, PFS and OS differences were found in an initial retrospective evaluation of efficacy as a function of morning vs. afternoon nivolumab administration in a first cohort of 57 patients [30]. We then prospectively assessed the nivolumab timing efficacy in a further 38 additional NSCLC patients in order to increase the sample size and provide independent confirmation of the initial findings. Here, we present the results of the whole study, which highlight a striking impact of nivolumab administration’s timing on its efficacy in 95 consecutive patients with metastatic NSCLC.

## 2. Patients and Methods

### 2.1. Patient Selection

For this study, we collected data from consecutive patients with histologically or cytologically documented locally advanced or metastatic NSCLC, treated with single agent nivolumab as second- or later-line treatment, at a single institution from September 2015 to November 2020. This retrospective evaluation of patients’ data according to nivolumab timing was approved by the local Institutional Review Board (#202028).

### 2.2. Nivolumab Treatment and Timing

Nivolumab was delivered intravenously every second week as a 1-h infusion at 3 mg/kg from September to 2015 to September 2018, then as a 30-min infusion of a fixed-dose of 240 mg, as recommended. All the patients started their treatment in the “day hospital” unit between 09:27 and 17:14. Nivolumab timing slots were randomly allocated for each course by the day-hospital coordinator on a logistical basis and accurately recorded. As a result, actual nivolumab timing could vary between courses for each patient, yet with a tendency to maintain similar time-slots for each patient for practical reasons. Thus, both the median time of nivolumab administration and its coefficient of variation were computed for each patient.

Patients received supportive care medications for immune-mediated adverse effects whenever indicated. Treatments were administered until disease progression, occurrence of unacceptable toxicities, or intercurrent death.

### 2.3. Assessments

The data collected included patient and tumor baseline characteristics, nivolumab’s actual dose and time of administration (date and clock hour), and the number of courses. WHO performance status was rated before the start of treatment. Anti-tumor responses were documented using thoraco-abdominopelvic computed tomography scans, cerebral magnetic resonance imaging and/or positron emission tomography with 18-fluorodeoxyglucose, which were performed after every four treatment courses. Responses were categorized according to iRECIST [31] criteria as complete (CR), partial (PR), stable (SD) or progressive disease (PD). Blood cell counts and differentials, renal and hepatic serum biochemistry, plasma thyroid hormones (TSH and fT4), and adverse events were assessed before each treatment course and graded according to NCI-CTCAE v4.0. All data were independently reviewed by two clinicians, alongside the lead oncologist for each patient, to ensure quality of data for consistency and plausibility.

### 2.4. Statistical Considerations

First, patients included in the study were dichotomized into two nivolumab timing groups relative to the median clock hour of all treatment courses. The ‘morning’ group included half of the patients who received the majority of the nivolumab administrations before this median clock hour, whereas the ‘afternoon’ group included the other half of the patients, for whom most of the nivolumab infusions were administered after the median time of all the courses. Patient characteristics, toxicities, and tumor response rates were compared using Fisher exact or Pearson χ^2^ tests. The intra-patient coefficient of variation was assessed by the ratio of the standard deviation to the mean of all infusion times for each individual patient. The median between-patients coefficient of variation of the nivolumab infusion times was then computed for each study group. PFS and OS were calculated from the date of the first administration of nivolumab until documented progression or death, respectively, and their curves were generated using the Kaplan–Meier method. The association between the nivolumab timing groups and PFS or OS was compared with the log-rank test.

Landmark PFS and OS analyses were performed, using 2 months for the base case analysis, then 4 months as representing the best balance allowing enough treatment exposure for responses to occur, whilst providing enough follow-up data to model survival beyond the landmark point. A landmark of 6 months was further explored for sensitivity analysis. Finally, multivariate Cox proportional hazards regression models were used to identify the main predictive or prognostic factors of PFS and OS with a minimum of 10 events per factor.

In a second sensitivity analysis, patients were also trichotomized in order to reduce the overlap that could result from the previous dichotomization: Group 1 received at least 67% of the infusions before the median clock hour of all treatment courses, Group 2 received at least 33% of the infusions before, and 33% of the infusions after this median time, and Group 3 involved those patients receiving at least 67% of the infusions after this median time. PFS and OS curves of the three groups were generated using the Kaplan–Meier method and compared with the log-rank test.

All analyses were performed with SPSS^®^ v26.0 software (Chicago, IL, USA). A *p*-value < 0.05 was considered statistically significant using bilateral tests.

## 3. Results

### 3.1. Patient Characteristics

Ninety-five consecutive patients were included in the study and categorized into two nivolumab timing groups according to the overall median treatment clock hour (Table 1). Thus, the majority of nivolumab administrations occurred between 9:27 and 12:54 for 48 patients (‘morning’ group) and between 12:55 and 17:14 for the 47 other patients (‘afternoon’ group) (Figure 1). The median nivolumab intra-patient timing coefficient of variation was 10%, ranging from 2% to 21%. The systematic determination of tumor molecular markers was recommended for routine oncology practice in 2020. Thus, mutations or rearrangements were searched for ALK1, ROS1 and EGFR, and PD-L1 expression was determined in the primary tumors of 43, 15, 32 and 72 patients, respectively. No mutation was found for ROS1. Mutations were identified for ALK1 in a single patient in the morning group, and in four patients for EGFR, including one with Exon 19 mutation and one with Exon 21 mutation in each group. Tumor PD-L1 expression exceeded 1% for 39 of the 72 patients tested (54.2%), with a similar distribution between the two nivolumab timing groups. A single patient who declined chemotherapy received nivolumab as a first-line treatment. Nivolumab was administered as a second-line treatment for 75.8% of the patients, and as third or later line for 23.2% of them, without any imbalance between the timing groups. Prior chemotherapy regimens included carboplatin (87.4% of the patients), cisplatin (17.9%), paclitaxel (69.5%), pemetrexed (33.7%), gemcitabine (20%), bevacizumab (10.5%), docetaxel (7.4%), erlotinib (4.2%), vinorelbine (2.1%), etoposide (1.1%) and osimertinib (1.1%). As compared to the patients in the ‘morning’ group, those in the afternoon group were slightly older (*p* = 0.012), more frequently female (*p* = 0.025), and tended to have a performance status of 1 rather than 0 (*p* = 0.061). No imbalance was apparent for any other patient or tumor characteristics (Table 1). A total of 1818 nivolumab treatment courses were administered with a median number of 9 courses per patient (range: 3 to 133). The median number of treatment courses was 19 (3 to 133) for the ‘morning’ group and 6 (3 to 30) for the ‘afternoon’ group (*p* < 0.001). The major reasons for treatment discontinuation irrespective of nivolumab timing were disease progression (74.7%), toxicity (7.4%), loss to follow-up (4.2%) and pulmonary sepsis (2.1%).

### 3.2. Treatment Safety

No toxic death was encountered. Fatigue was the most frequent grade 3 or 4 adverse event. It occurred in 10.5% of the patients, including three patients in the ‘morning’ group (6.3%) and seven patients in the ‘afternoon’ group (14.9%) (*p* = 0.024) (Table 2). In contrast, grade 2–3 skin toxicities occurred more frequently in the ’morning’ group (31.9% vs. 12.7%, *p* = 0.049). Toxicities were otherwise infrequent and similar in both treatment timing groups. Intercurrent events with questionable relation to nivolumab treatment were also observed, including pulmonary sepsis (38 patients, 40%), pulmonary embolism (3 patients, 3.2%) and thrombosis (6 patients, 6.4%). The frequency of these intercurrent events was similar in both groups except for pulmonary sepsis (‘morning’ and ‘afternoon’ groups, 17.9% and 7.4%, respectively, *p* = 0.049).

### 3.3. Tumor Responses

The rates of objective responses (ORR) and disease control (DCR) were 27.7% (95% CL: 18.7 to 36.7) and 63.8% (54.1 to 73.5), respectively, for the 95 patients. ORR and DCR were significantly larger in the ‘morning’ group as compared to the ‘afternoon’ group (ORR: 37.5% vs. 14.4%, *p* = 0.029; DCR: 77.1% vs. 50.0%, *p* = 0.006).

### 3.4. Progression-Free Survival (PFS) and Overall Survival According to Nivolumab Timing

In the 95 patients, median PFS and OS (95% CL) were 3.9 months (2.1–5.8) and 14.0 months (9.5–18.4), respectively. Strikingly, median PFS and median OS were both nearly four times as long following ‘morning’ rather than ‘afternoon’ nivolumab dosing (Figure 2). Median PFS varied from 11.3 months (5.5–17.1) for the ‘morning’ group down to 3.1 months (1.5–4.6) for the ‘afternoon’ group (*p* < 0.001) (Figure 2a). Corresponding median OS values were 34.2 months (15.1–53.3) and 9.6 months (4.9–14.4), respectively (*p* < 0.001 (Figure 2b).

Landmark analysis revealed that both PFS and OS were significantly longer for the ‘morning’ vs. ‘afternoon’ groups already at 2 months. Thus, the median PFS in the 42 patients in the ‘morning’ group exposed to nivolumab at the 2-month landmark was 15.8 months (6.2–25.5), whereas it was 3.5 months (3.2–3.9) for the 31 patients in the ‘afternoon’ group with the same nivolumab exposure (*p* < 0.001) (Figure 3). Similar results were obtained at the 4-month and 6-month landmark analyses (Figure 3). These findings indicated that nivolumab timing was critical for efficacy from the very beginning of treatment administration, and that the duration of tumor control by immunotherapy was critical for OS.

### 3.5. Did Patient Performance Status Influence Nivolumab Timing-Dependent Efficacy?

Median PFS was significantly longer in the 35 patients with a PS of 0 (11.1 months (2.2–20.0)) as compared to that in the 56 patients with PS = 1 (3.3 months (2.4–4.2)) (*p* = 0.020). A similar trend was found for OS according to PS, although non-significantly so (PS = 0: 19.5 months (8.2–30.8); PS = 1: 11.8 months (9.4–14.2); *p* = 0.13). Notwithstanding, ‘morning’ nivolumab significantly improved both PFS and OS in each PS category in comparison with ‘afternoon’ administration (Figure 4a,b).

### 3.6. Did Tumor PD-L1 Expression Influence Nivolumab Timing-Dependent Efficacy?

The 39 patients with PD-L1-positive (+) NSCLC had longer PFS and OS as compared to the 33 patients with PD-L1-negative (-) tumors. However, the differences were not statistically significant either for PFS (PD-L1(+): 9.6 months (2.4–16.8); PD-L1(-): 3.4 months (0.0–6.8); *p* = 0.071) or for OS (PD-L1(+): 22.2 months (7.0–37.4); PD-L1(-): 12.4 months (5.5–19.3); *p* = 0.57).

Most importantly though, ‘morning’ nivolumab nearly quadrupled both median PFS and median OS as compared to ‘afternoon’ dosing for patients in each tumor PD-L1 category (Figure 4c,d).

### 3.7. Univariate and Multivariate Cox Model Analyses of PFS and OS

‘Morning’ nivolumab consistently reduced the respective risks of an earlier progression or death as compared to ‘afternoon’ dosing in each prognostic category according to the Cox proportional univariate hazard ratios (Figure 5). Multivariate analyses further confirmed nivolumab timing as an independent prognostic factor for both PFS and OS (Table 3). The best outcomes resulted from ‘morning’ administration, with hazard ratios (HR) of 0.258 (0.115–0.580) for PFS and of 0.174 (0.082–0.370) for OS (*p* < 0.001 for both). Another joint significant prognostic factor was treatment line, with second-line nivolumab being associated with better PFS and OS as compared to third or later line. No other prognostic factors were identified for PFS. For OS, PS, histological type as well as bone and liver metastases were also statistically significant, with improved OS being predicted by a PS of 0, by the primary tumor being an adenocarcinoma rather than a squamous cell carcinoma, and by the lack of bone or liver metastases.

### 3.8. Nivolumab’s Efficacy According to Infusion Time following Trichotomization of Patients

There were 36 patients receiving at least 67% of the infusion before 12:54 (Group 1), 24 patients receiving at least 33% of the infusion before 12:54 and 33% after 12:54 (Group 2) and 35 patients receiving at least 67% of the infusion after 12:54 (Group 3) (Appendix A). Patient characteristics did not differ significantly between groups, except for liver metastases, which were predominant in group 1 (*p* = 0.010).

Efficacy results revealed consistent and statistically significant differences for both PFS and OS among the three groups (*p* = 0.002 and *p*= 0.003, respectively) (Figure 6). Most importantly, consistent trends were found with Group 1 displaying longest median values for PFS (11.1 (0.0–25.5) and for OS (34.2 (NR, NR)), Group 2 having intermediate median values (PFS: 5.9 (0.0–13.3); OS: 15.3 (8.0–22.7)) and Group 3 experiencing the worst outcomes (median PFS: 3.1 (1.6–4.5); median OS: 12.4 (4.0–20.7)). These results supported the relevance of morning nivolumab dosing, and further highlighted that poor efficacy was associated with more infusions being delivered after 12:54.

At the 2-month landmark, 73 patients (76.8%) were exposed to nivolumab treatment, including 42 in the ‘morning’ group and 31 in the ‘afternoon’ group. The four-month landmark concerned 50 patients (52.6%), 32 and 18 in the ‘morning’ and the ‘afternoon’ groups, respectively. Finally, 37 patients were exposed at the 6-month landmark point, 27 patients in the ‘morning’ group and 10 patients in the ‘afternoon’ group.

Median PFS at the 2-, 4- and 6-month landmark was 15.8 months (6.2–25.5), 20.1 (9.7–30.6) and 25.9 (16.8–35.0) in the ‘morning’ group, respectively, and 8.6 (3.2–3.9), 8.6 (5.0–12.2) and 9.7 (8.0–11.4) in the ‘afternoon’ group, respectively. PFS was statistically different at each of the three landmark points with respective *p*-values of <0.001, 0.001 and 0.005.

Median OS at the 2-, 4- and 6-month landmark was 38.2 months (22.4–53.9), 44.5 (31.8–57.1) and 44.5 (32.9–56.1), respectively, in the ‘morning’ group and 12.4 (7.4–17.3), 12.4 (10.4–14.4) and 15.3 (8.0–22.7), respectively, in the ‘afternoon’ group. *p*-values were < 0.001 at each of the three landmark points.

## 4. Discussion

To the best of our knowledge, no study has yet explored whether the daily timing of anti-PD-1/PD-L1 immune checkpoint inhibitor drugs could moderate therapeutic efficacy in NSCLC patients. Our current investigation revealed for the first time that both median PFS and median OS were nearly quadrupled in metastatic NSCLC receiving the majority of nivolumab administrations between 9:27 and 12:54, rather than between 12:55 and 17:14. The large difference in both endpoints was consistent in all patient subgroups, with similar hazard ratios for the reduction in an earlier progression or an earlier death. Additionally, nivolumab’s timing of administration was confirmed as an independent prognostic indicator of both PFS and OS, with hazard ratios of 0.258 and 0.174 in respective multivariate analyses. These results were consistent in all the subgroup analyses performed. Moreover, the trichotomization of the patient population further showed that the best outcomes resulted from delivering 67% of nivolumab infusion before 12:54. Taken together, the results strongly support a dramatic increase in nivolumab’s efficacy following morning dosing.

The main limitation of our study is related to its retrospective nature and to the lack of statistical randomization between ‘morning’ and ‘afternoon’ nivolumab administrations. However, nivolumab timing was allocated for each course in each patient by the day-hospital coordinator, on a strictly logistical basis, thus reducing the risk of an uncontrolled bias. Moreover, both PFS and OS in our whole study population of 95 consecutive patients were comparable to the landmark trials with NSCLC patients receiving time-unspecified nivolumab [32,33,34,35]. Some studies have demonstrated improved efficacy of nivolumab in NSCLC patients whose tumors expressed PD-L1 [4,36,37]. We confirmed such a trend for PFS, but not for OS, in our study patients, possibly because of the limited size of our study population, using the standard cut-off of 1% of PD-L1-positive cells for considering a tumor as being PD-L1(+) [4]. Strikingly however, nivolumab’s timing of administration significantly influenced PFS and OS both in the patients with PD-L1(+) tumors and in those with PD-L1(-) tumors. Similarly, ‘morning’ nivolumab also significantly improved PFS and OS in patients with a PS of 0 and in those with a PS of 1 in comparison with ‘afternoon’ dosing.

Skin toxicities were reported to predict for improved PFS and OS on nivolumab [38]. We confirmed this finding here as well, with 29.8% patients with grade 2 skin toxicities on morning dosing as compared to 10.6% of those on afternoon dosing. Consistent, with other reports [39,40], grade 3–4 fatigue was the main adverse event, yet it occurred in 6.3% of the patients on ‘morning’ nivolumab as compared to 14.9% following ‘afternoon’ dosing.

The landmark analyses of PFS and OS according to nivolumab timing further revealed that the timing-related differences were detected from the first 2 months on treatment, thus indicating that nivolumab’s effects might be impaired in the ‘afternoon’ from the very beginning of such immunotherapy. Indeed, the major cause of treatment discontinuation was progressive disease, which occurred very early in the ‘afternoon’ group, resulting in fewer courses being given to the patients treated predominantly later in the day.

Our results are in rather good agreement with those recently reported in patients with metastatic malignant melanoma by Qian DC et al. [38], although they categorized their patients differently. More precisely, they used 16:30 as a temporal cut-off, and showed that the administration of 20% or more of ICI’s infusions later than 16:30 increased the relative risk of an earlier death with an HR of 2.04 (1.04–4.0) (*p* = 0.038) in a propensity score-matched analysis involving 146 patients.

Taken together, both study results indicate that the biological mechanisms of nivolumab and other ICIs could differ largely according to their dosing time. Indeed, differences in time-dependent tolerability and/or efficacy have been shown for nearly 50 anticancer drugs in experimental models and/or cancer patients, with the main mechanisms involving circadian changes in pharmacokinetics and pharmacodynamics [21,28,41,42,43]. Understanding the main mechanisms at work for chronotherapy with nivolumab and other ICIs represents an important challenge. The average half-life of nivolumab of 20 days, with a reported sustained mean occupancy of >70% of PD-1 molecules on circulating T cells for two months or more following infusion [44], indeed suggests a limited role for circadian time-dependent pharmacokinetics as the main mechanism of its chrono-efficacy.

However, we believe that our understanding of nivolumab’s binding dynamics, likely tested at a single time of day, is probably not correct. Rather, our data and those of Qian et al. invite a more fundamental scientific look at the rhythmic binding over 24 h, as well as the trafficking of T-cells into the tumor-draining lymph nodes. Thus, whilst PD-L1 expression on tumor cells has been considered as a critical mechanism of nivolumab’s anti-tumor activity, this factor was not proven as an essential determinant of clinical efficacy in NSCLC. Although the selection of high tumor PD-L1 expression subsets enriched the response to anti–PD-1 therapy, more than half of these patients lacked durable responses to PD-1 blockade [45]. Such results concurred with the lack of survival prediction by PD-L1 positivity in our study.

Large amplitude circadian rhythms in T- and B-lymphocytes trafficking and functions have long been known, both in rodents and in humans. For instance, the counts of circulating total T lymphocytes and their subsets were nearly twice as high in healthy people at 08:00 in the morning as compared to midnight or 04:00 at night [15,46,47]. An experimental study further revealed a precise circadian control of lymphocyte trafficking from tissues to blood and vice-versa [46], which might be relevant here. Most importantly, no homing of T-lymphocytes from blood to lymph nodes was demonstrated in the second half of the nocturnal activity span of mice [48], which would correspond to afternoon and evening in humans, and might account for the observed timing-related differences in nivolumab’s efficacy. Furthermore, a circadian rhythm characterized the in vivo T-(CD8+) cell response to immunization of mice with dendritic cells loaded with ovalbumin antigen [14,49]. In humans, BCG vaccination in the morning induced stronger trained immunity and adaptive responses compared with evening vaccination [12]. Increased antibody responses were also reported for morning versus afternoon vaccinations against influenza virus [50]. The reduced activity of afternoon nivolumab could involve the exacerbation of resistance mechanisms in this temporal window, including defective antigen presentation processes, multiple immune checkpoint interactions, transient refractoriness of the tumoral immune microenvironment, activation of oncogenic pathways, and epigenetic changes of key proteins in tumors, among other factors [51,52]. Nivolumab could also become inactivated shortly after its infusion through binding to circulating proteins or extracellular vesicles [53,54], thus accounting for consistent resistance in the ‘afternoon’ group in the landmark analyses. In general, the relevance of dosing time for cancer treatments has been underestimated in oncology, with temporal logistics organization of hospitals predominating over temporal changes in biological functions. Education of the medical community to circadian clocks and their relevance for health and treatment effects has been largely lacking. Finally, the drug companies that have developed immune checkpoint inhibitors have not investigated whether timing could impact efficacy (or have not reported their results on this issue). The lack of mandatory investigations on drug timing before drug registration (or in postmarketing studies) is also a critical consideration that lets people believe that immunotherapy timing would have no real effect. Our results, and those of Qian et al. [26], clearly highlight that this bottleneck can and needs to be further circumvented through the undertaking of randomized clinical trials.

## 5. Conclusions

In conclusion, we feel that the fourfold difference in overall survival found in NSCLC patients according to nivolumab circadian timing supports the morning administration of this checkpoint inhibitor at arguably no cost. Our results, however, highlight the need for prospective randomized-controlled timing studies of nivolumab as well as of other anticancer immunotherapies. Carefully conducted translational investigations are mandatory to precisely identify the clinico-biological mechanisms underlying the major circadian dependencies in nivolumab’s efficacy, so as to optimize individual patient benefits on immunotherapy.

## Figures and Tables

**Figure 1 cancers-14-00896-f001:**
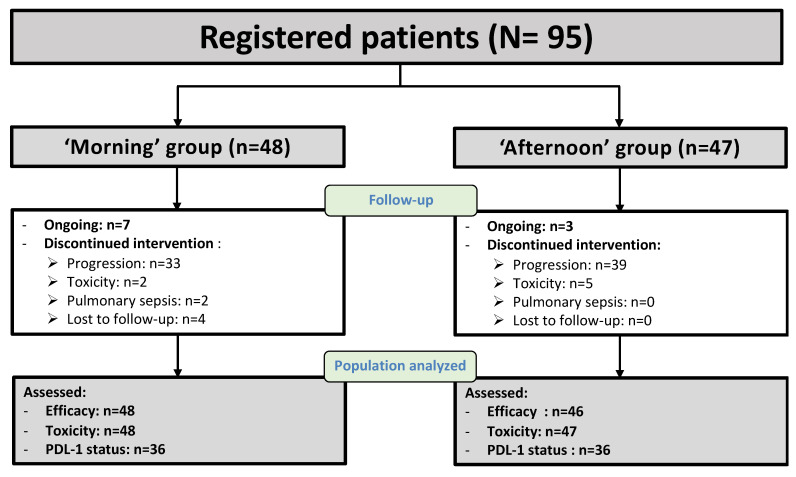
Study flow diagram.

**Figure 2 cancers-14-00896-f002:**
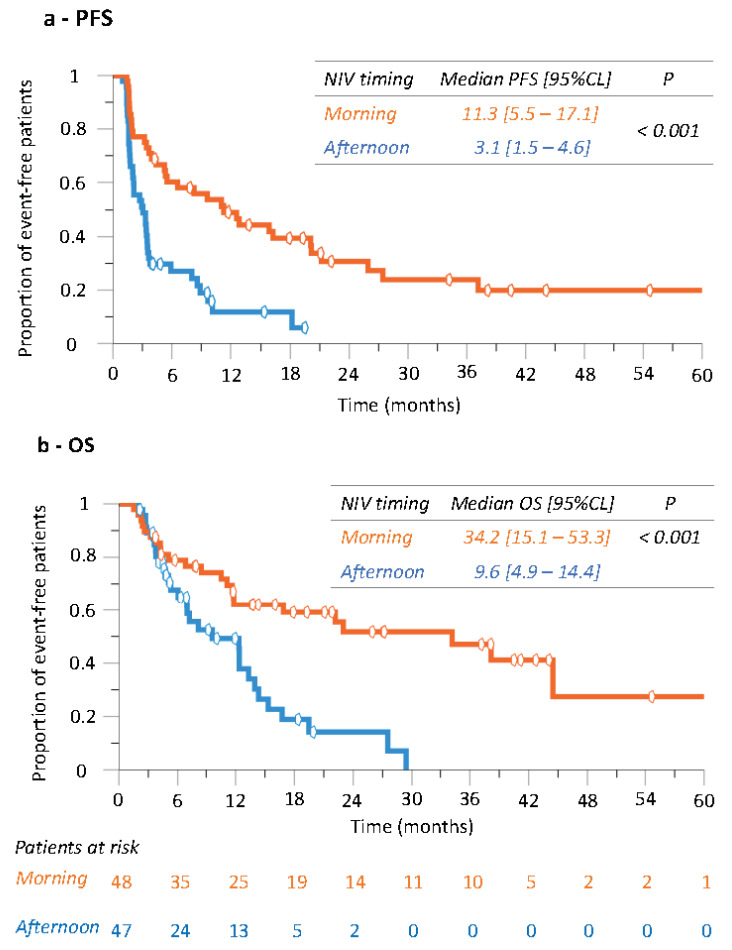
Kaplan–Meier curves of progression-free survival (PFS, (**a**)) and overall survival (OS, (**b**)) according to nivolumab timing group in whole study population.

**Figure 3 cancers-14-00896-f003:**
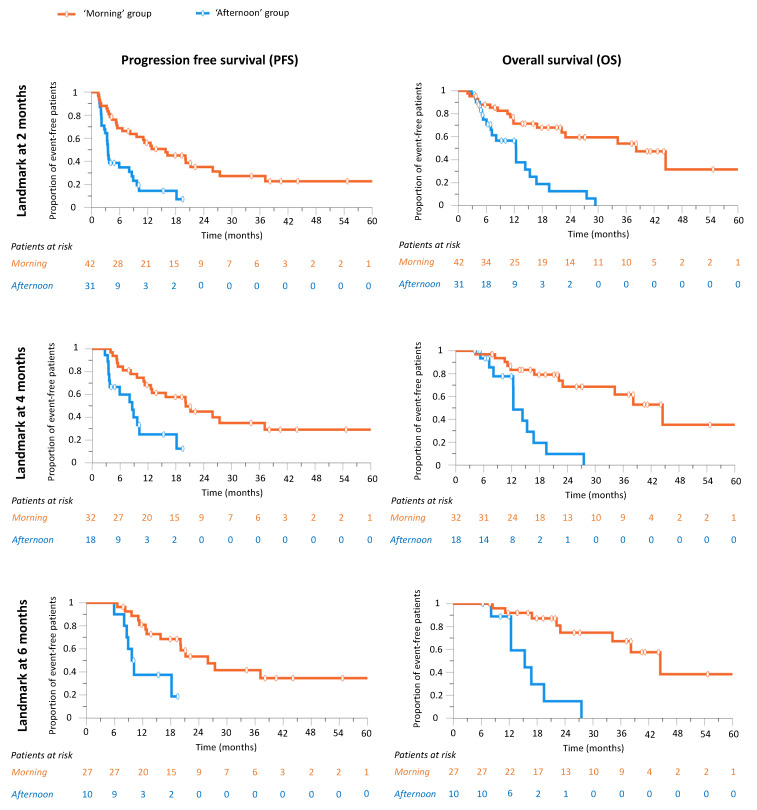
Progression-free survival (PFS) and overall survival (OS) landmark analysis by nivolumab timing group at 2, 4 and 6 months of exposure to nivolumab.

**Figure 4 cancers-14-00896-f004:**
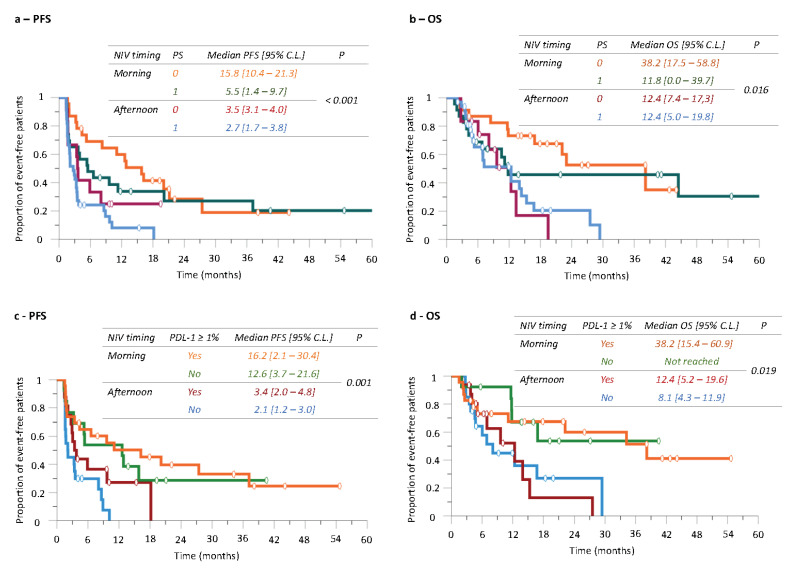
Kaplan–Meier curves of progression-free survival (PFS) and overall survival (OS) according to nivolumab timing group and WHO PS (**a**,**b**) or PD-L1 status (**c**,**d**).

**Figure 5 cancers-14-00896-f005:**
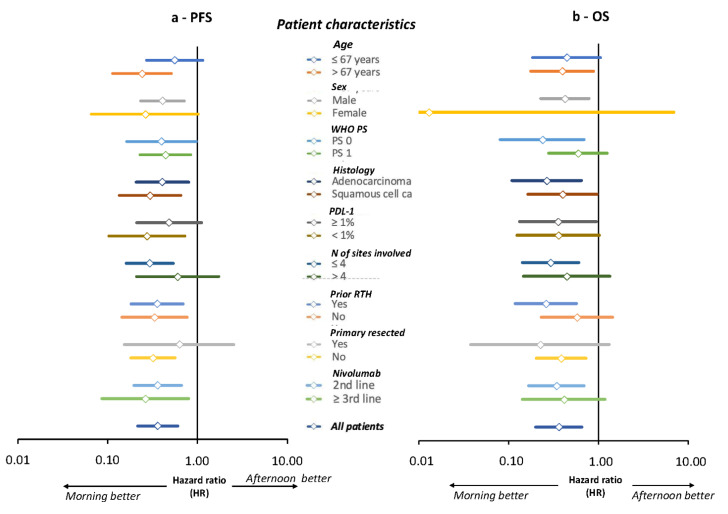
Forest plots of progression-free survival (PFS, (**a**)) and overall survival (OS, (**b**)), according to categorized patient characteristics. Hazard ratios and 95% CL of an earlier progression or an earlier death for ‘morning’ vs. ‘afternoon’ nivolumab groups.

**Figure 6 cancers-14-00896-f006:**
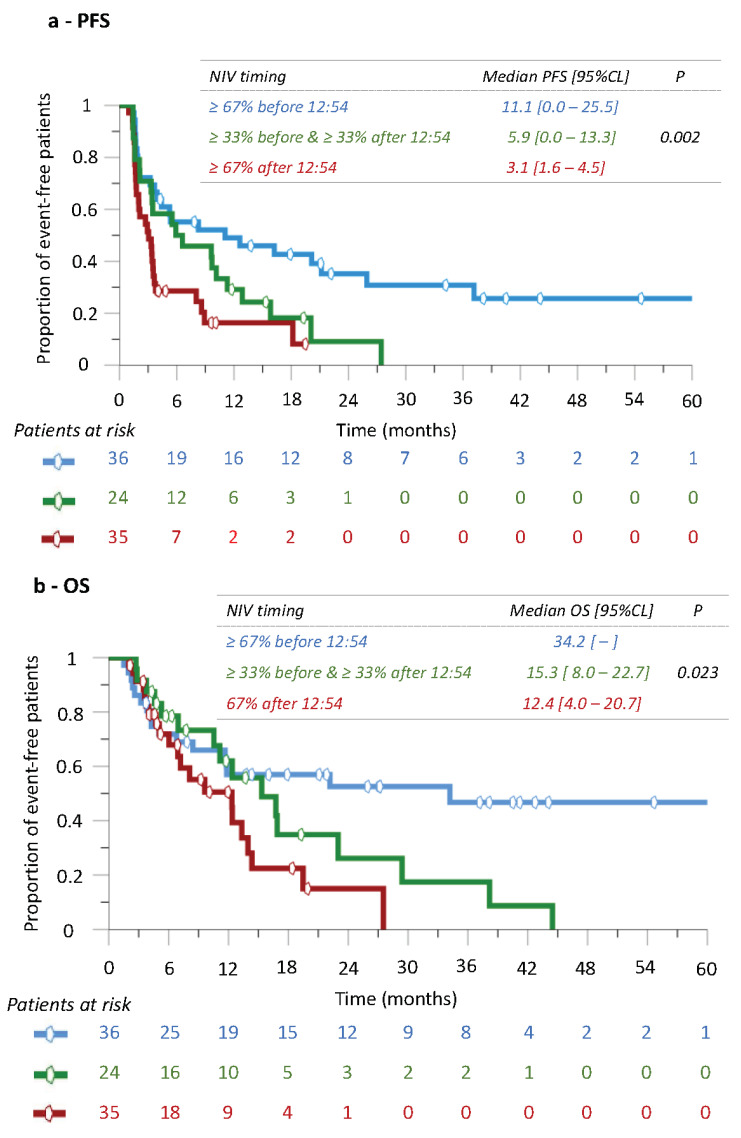
Kaplan–Meier curves of progression-free survival (**a**) and overall survival (**b**), according to three timing groups. Group 1 received at least 67% of the infusions before 12:54, Group 2 received at least 33% of the infusions before 12:54 and 33% of the infusions after 12:54, and Group 3 involved those patients receiving at least 67% of the infusions after 12:54.

**Table 1 cancers-14-00896-t001:** Main baseline characteristics of whole study population, and according to nivolumab (NIV) timing groups.

Characteristics	All Patients (N = 95)	‘Morning’ Group (n = 48)	‘Afternoon’ Group (n = 47)	*p*-Value
**Timing of >50% of infusions**		**09:27 to 12:54**	**12:55 to 17:14**	
**Actual NIV ^a^ timing**				
Median time, hh:min	12:54	11:55	13:34	**<0.001**
(range)	(9:27–17:14)	(9:27–16:52)	(10:10–17:14)	
(IQR)	(11:55–13:34)	(11:11–12:26)	(13:10–14:44)	
Median intra-patient CV ^b^	10%	11%	8%	
(range)	(2%–21%)	(2%–21%)	(2%–19%)	
**Age, years**				
Median (range)	66.9 (41.2–82.5)	66.2 (41.2–80.8)	69.2 (48.8–82.5)	**0.012**
**Sex**				
Female	16 (16.8%)	4 (8.3%)	12 (25.5%)	**0.025**
Male	79 (83.2%)	44 (91.7%)	35 (74.5%)	
**Tobacco-induced COPD ^c^**	74 (77.9%)	37 (77.1%)	37 (78.7%)	0.847
**Histological type**				
Adenocarcinoma	55 (57.9%)	26 (54.2%)	29 (61.7%)	0.696
Squamous cell carcinoma	37 (38.9%)	20 (41.7%)	17 (36.2%)	
NSCLC unspecified	3 (3.2%)	2 (4.2%)	1 (2.1%)	
**Tumor PD-L1 expression**				0.098
≥1%	39 (41.1%)	23 (47.9%)	16 (34.0%)	
<1%	33 (34.7%)	13 (27.1%)	20 (42.6%)	
Not assessed	23 (24.2%)	12 (25.0%)	11 (23.4%)	
**Primary resected**	15 (15.8%)	9 (18.8%)	6 (12.8%)	0.424
**Prior adjuvant chemo.**	11 (11.6%)	7 (14.6%)	4 (8.5%)	0.355
**Prior radiotherapy**	57 (60.0%)	27 (56.3%)	30 (63.8%)	0.451
**N of prior chemo. lines**				
0	1 (1.1%)	1 (2.1%)	0 (0.0%)	0.955
1	72 (75.8%)	36 (75.0%)	36 (76.6%)	
2–5	22 (23.2%)	11 (22.9%)	11 (23.4%)	
**Number of sites involved**				
Median (range)	4 (1–8)	4 (1–7)	3 (2–8)	0.854
**Site of metastases**				
Brain	23 (24.2%)	12 (25.0%)	11 (23.4%)	0.856
Pleura	39 (41.1%)	16 (33.3%)	23 (48.9%)	0.122
Bone	49 (51.6%)	21 (43.8%)	28 (59.6%)	0.123
Liver	24 (25.3%)	15 (31.3%)	9 (19.1%)	0.175
Adrenal gland	19 (20.0%)	10 (20.8%)	9 (19.1%)	0.837
Pericardium	7 (7.4%)	4 (8.3%)	3 (6.4%)	1
**WHO performance status**				
0	35 (36.8%)	23 (47.9%)	12 (25.5%)	0.061
1	56 (58.9%)	23 (47.9%)	33 (70.2%)	
2	4 (4.2%)	2 (4.2%)	2 (4.3%)	

*^a^ Nivolumab; ^b^ coefficient of variation; ^c^ chronic obstructive pulmonary disease.*

**Table 2 cancers-14-00896-t002:** Toxicity grades in all 95 patients and separately in ‘morning’ and ‘afternoon groups.

Toxicities	Toxicity Grade	All (n = 95)	Morning’ Group (n = 48)	Afternoon’ Group (n = 47)	*p*-Value
**Fatigue**	*Grade 0*	43 (45.3%)	20 (41.7%)	23 (48.9%)	**0.024**
	*Grade 1*	13 (13.7%)	11 (22.9%)	2 (4.3%)	
	*Grade 2*	29 (30.5%)	14 (29.2%)	15 (31.9%)	
	*Grade 3*	9 (9.5%)	2 (4.2%)	7 (14.9%)	
	*Grade 4*	1 (1.1%)	1 (2.1%)	0 (0.0%)	
**Skin toxicity**	*Grade 0*	70 (74.5%)	30 (63.8%)	40 (85.1%)	**0.049**
	*Grade 1*	3 (3.2%)	2 (4.3%)	1 (2.1%)	
	*Grade 2*	19 (20.2%)	14 (29.8%)	5 (10.6%)	
	*Grade 3*	2 (2.1%)	1 (2.1%)	1 (2.1%)	
**Anorexia**	*Grade 0*	71 (74.7%)	36 (75.0%)	35 (74.5%)	0.856
	*Grade 1*	11 (11.6%)	6 (12.5%)	5 (10.6%)	
	*Grade 2*	9 (9.5%)	5 (10.4%)	4 (8.5%)	
	*Grade 3*	4 (4.2%)	1 (2.1%)	3 (6.4%)	
**Anemia**	*Grade 0*	72 (75.8%)	39 (81.3%)	33 (70.2%)	0.193
	*Grade 1*	6 (6.3%)	1 (2.1%)	5 (10.6%)	
	*Grade 2*	16 (16.8%)	7 (14.6%)	9 (19.1%)	
	*Grade 3*	1 (1.1%)	1 (2.1%)	0 (0%)	
**Nausea**	*Grade 0*	75 (78.9%)	40 (83.3%)	35 (74.5%)	0.633
	*Grade 1*	15 (15.8%)	6 (12.5%)	9 (19.1%)	
	*Grade 2*	4 (4.2%)	2 (4.2%)	2 (4.3%)	
	*Grade 4*	1 (1.1%)	0 (0.0%)	1 (2.1%)	
**Hepatitis**	*Grade 0*	80 (84.2%)	39 (81.3%)	41 (87.2%)	0.828
	*Grade 1*	13 (13.7%)	7 (14.6%)	6 (12.8%)	
	*Grade 2*	1 (1.1%)	1 (2.1%)	0 (0%)	
	*Grade 3*	1 (1.1%)	1 (2.1%)	0 (0.0%)	
**Renal failure**	*Grade 0*	80 (84.2%)	38 (79.2%)	42 (89.4%)	0.173
	*Grade 1*	15 (15.8%)	10 (20.8%)	5 (10.6%)	
**Vomiting**	*Grade 0*	81 (85.3%)	42 (87.5%)	39 (83.0%)	0.75
	*Grade 1*	9 (9.5%)	4 (8.3%)	5 (10.6%)	
	*Grade 2*	3 (3.2%)	1 (2.1%)	2 (4.3%)	
	*Grade 3*	1 (1.1%)	1 (2.1%)	0 (0.0%)	
	*Grade 4*	1 (1.1%)	0 (0.0%)	1 (2.1%)	
**Hypothyroidism**	*Grade 0*	83 (87.4%)	40 (83.3%)	43 (91.5%)	0.353
	*Grade 1*	8 (8.4%)	6 (12.5%)	2 (4.3%)	
	*Grade 2*	4 (4.2%)	2 (4.2%)	2 (4.3%)	
**Muscle toxicity**	*Grade 0*	83 (87.4%)	41 (85.4%)	42 (89.4%)	0.102
	*Grade 1*	4 (4.2%)	4 (8.3%)	0 (0.0%)	
	*Grade 2*	6 (6.3%)	3 (6.3%)	3 (6.4%)	
	*Grade 3*	2 (2.1%)	0 (0.0%)	2 (4.3%)	
**Diarrhea**	*Grade 0*	84 (88.4%)	43 (89.6%)	41 (87.2%)	0.197
	*Grade 1*	6 (6.3%)	4 (8.3%)	2 (4.3%)	
	*Grade 2*	3 (3.2%)	0 (0.0%)	3 (6.4%)	
	*Grade 3*	1 (1.1%)	1 (2.1%)	0 (0.0%)	
	*Grade 4*	1 (1.1%)	0 (0.0%)	1 (2.1%)	
**Abdominal pain**	*Grade 0*	87 (91.6%)	43 (89.6%)	44 (93.6%)	0.24
	*Grade 1*	3 (3.2%)	2 (4.2%)	1 (2.1%)	
	*Grade 2*	3 (3.2%)	3 (6.3%)	0 (0.0%)	
	*Grade 3*	2 (2.1%)	0 (0.0%)	2 (4.3%)	

**Table 3 cancers-14-00896-t003:** Prognostic factors of PFS and OS.

	**PFS**			
			**95% CL of hazard ratio**	
**Prognostic factors of PFS**	***p*-values**	**Hazard ratio**	**CL inf**	**CL sup**
NIV dosing time	0.001	0.258	0.115	0.580
2nd line NIV	0.009	0.324	0.140	0.752
	**OS**			
			**95% CL of hazard ratio**	
**Prognostic factors of OS**	***p*-value**	**Hazard ratio**	**CL inf**	**CL sup**
NIV dosing time	<0.001	0.174	0.082	0.370
2nd line NIV	0.004	0.390	0.204	0.746
Adenocarcinoma	<0.014	0.460	0.248	0.854
Bone not involved	0.005	0.420	0.230	0.765
Liver not involved	<0.001	0.300	0.153	0.588
WHO PS 0	0.048	2.005	1.007	3.993

## Data Availability

The data presented in this study are available on request from the corresponding author.

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
