# Peer review of "Time-Dependent Efficacy of Checkpoint Inhibitor Nivolumab: Results from a Pilot Study in Patients with Metastatic Non-Small-Cell Lung Cancer"

_cancers, 2022, doi:10.3390/cancers14040896_

Round 1

Reviewer 1 Report

As recognized by the authors (rows 337-338) the study has relevant limitations related to its design (initial retrospective evaluation) and lack of statistical randomization, whereby only a well-designed randomized trial could confirm the proof of concept.

In any case, in routine clinical practice nivolumab is generally administered in the morning in oncological day hospitals.

The revised version reported data of a recent paper about chronoimmunotherapy published by Quian et al., which is however the only one present in the literature so far.

Author Response

As recognized by the authors (rows 337-338) the study has relevant limitations related to its design (initial retrospective evaluation) and lack of statistical randomization, whereby only a well-designed randomized trial could confirm the proof of concept.

Response: We agree with this comment. Our current pilot study provides a proof of concept that nivolumab timing matters for NSCLC patients. The quantitative estimates of dosing time-related differences, that did not exist before, are guiding us for the design of a randomized trial, which is our next step.

In any case, in routine clinical practice nivolumab is generally administered in the morning in oncological day hospitals.

Response: Morning dosing of nivolumab may be the rule in the practices known to the reviewer, and this is really fortunate for the patients, in view of our results and those of Quian et al. Unfortunately, this is not the case everywhere in the world. Thus, Qian et al report immune checkpoint inhibitors (ICI) infusion times ranging from 08:00 to 17:30 at Emory University Hosp (Atlanta, GE) (see appendix page 2 in their manuscript). In our study conducted in the North of Paris area, the timing of nivolumab infusions ranged from 9:27 To 17:14 (Table 1). We currently have access to other databases from other cancer centers in France, which reveal similar data, i.e. the ranges of ICI’s infusions times indeed range over an~ 8-h range. So the evidence regarding timing of nivolumab and other ICI’s is highlighting large variations in dosing times between medical centers.

The revised version reported data of a recent paper about chronoimmunotherapy published by Quian et al., which is however the only one present in the literature so far.

Response: We agree with this comment. We are proud of being the first ones to report similar findings in another cancer type, i.e. non small cell lung cancer rather than malignant melanoma. We are aware of many groups working on this issue, which has now become highly competitive.

Reviewer 2 Report

The manuscript is acceptable in its current form. Also, based on the revisions that were made.

Reviewer 3 Report

Authors did great job and it worth appreciating. The paper has several questions.

First, please explain the age difference between study groups in Table 1. How many patients were older than 60 y.o.?

Second, please explain received  median nivolumab intra-patient timing coefficient of variation.

Third, explain Kaplan-Meier curves of progression free survival and overall survival Kaplan-Meier curves of progression free survival according to nivolumab timing. I recommend to get pictures of better quailty 

Authors stated that  PD-1 factor was not proven as an essential determinant of clinical efficacy in 
NSCLC. Please propose the possible mechanistic explanation.

Author Response

Authors did great job and it worth appreciating. The paper has several questions.

First, please explain the age difference between study groups in Table 1. How many patients were older than 60 y.o.?

Response: In our study, 73 patients (76.8%) were older than 60 y.o, including 33 (68.7%) in the ‘morning’ group, and 40 (85.1%) in the ‘afternoon’ group. The imbalance in age distribution between both groups is due to the retrospective nature of our study, with no possible stratification by age, since we elected to evaluate the 95 consecutive patients that presented to our clinics for receiving nivolumab for NSCLC. Treatment administration times were decided by the hospital day coordinator, according to the availability of time slots for patients’ treatments, i.e. dosing times were allocated according to logistic bases, without any consideration for the patient’s or the doctor’s preference, as reported in the Methods.  The imbalance in age between both timing groups did not influence the relevance of nivolumab timing for reducing the risk of an earlier progression or an earlier death, as shown in the calculations of the Hazard Ratios according to age for both efficacy endpoints (Figure 5). Thus, the benefit from ‘morning’ dosing was similar in the patients aged < or > 67 y.o. (the median age of all the patients). Moreover, age was not a significant predictor of PFS or OS in multivariable analyses (Table 3).

Second, please explain received  median nivolumab intra-patient timing coefficient of variation.

Response: Nivolumab timing coefficient of variation reflected the relative variability of the administration times of nivolumab both within and between patients. The intra-patient coefficient of variation was assessed by the ratio of the standard deviation to the mean of all infusion times for each individual patient. The median between-patients coefficient of variation of the nivolumab infusion times was then computed for each study group.

We have added the following sentence in the Methods:

“The intra-patient coefficient of variation was assessed by the ratio of the standard deviation to the mean of all infusion times for each individual patient. The median between-patients coefficient of variation of the nivolumab infusion times was then computed for each study group”.

Third, explain Kaplan-Meier curves of progression free survival and overall survival Kaplan-Meier curves of progression free survival according to nivolumab timing. I recommend to get pictures of better quality.

Response: Kaplan-Meier method was used to plot progression-free survival and overall survival curves. For each survival category, the ‘morning’ and ‘afternoon’ groups were plotted separately. The previous figures 2, 3, 4 and 6 have been hopefully improved through increasing the thickness of the lines, among other corrections.

Authors stated that  PD-1 factor was not proven as an essential determinant of clinical efficacy in NSCLC. Please propose the possible mechanistic explanation.

Response: To our knowledge, PD-1 has not been studied as a predictor of effectiveness of checkpoint inhibitors in NSCLC. In contrast, tumor PD-L1 expression predicted for improved efficacy of immune checkpoint inhibitors in NSCLC [cited  ref 1-6 ]. However, not all NSCLC patients with high PD-L1 expression responded to PD-1/PD-L1 blockade, and patients with PD-L1 negative tumors also responded significantly [1-6]. In view of these facts, it can be assumed that PD-L1 is not the single  determinant of the clinical efficacy of immune checkpoint inhibitors in NSCLC. Thus, we believe that host immune factors including tumor infiltration by dendritic cells  and lymphocytes, and activated T-lymphocytes in tumor draining lymph nodes  may play a major role as already mentioned in our manuscript (Lines 430 -434 [cited ref 51, 52]. The results of our pilot study point in the same direction with the time of administration of nivolumab being the major determinant factor of progression-free survival and overall survival irrespective of PD-L1 status. However PD-L1 positivity was associated with PFS prolongation in both timing groups, yet, it had no influence on overall survival.

Reviewer 4 Report

The authors responded to the comments of the reviewer and made significant changes to the text of the manuscript. However, I still have questions. 1. In the morning group, 23 patients expressed PD-L1 and 13 did not; in the afternoon group, 16 expressed PD-L1 and 20 did not. It is only on this basis that one should expect a greater response to treatment in the first group. Do you agree with that? 2. I am confused by the accuracy of determining the time boundary 12:54. By this logic, there will be a difference in outcome between patients receiving chemotherapy at 12-53 and 12-55. 3. Given the significant difference in survival rates, suspiciously few scientists have investigated this issue in the literature. What is the reason for this in your opinion?

Author Response

The authors responded to the comments of the reviewer and made significant changes to the text of the manuscript. However, I still have questions.

  1. In the morning group, 23 patients expressed PD-L1 and 13 did not; in the afternoon group, 16 expressed PD-L1 and 20 did not. It is only on this basis that one should expect a greater response to treatment in the first group. Do you agree with that?

Response:

No, because PDL-1 played no role on nivolumab efficacy both irrespectively of timing and each timing group (morning or afternoon) as showed by univariate HR of timing effect. In both PDL-1 positive and PDL-1 negative groups, PFS and OS were significatively better for the morning group (Figure 5).

  1. I am confused by the accuracy of determining the time boundary 12:54. By this logic, there will be a difference in outcome between patients receiving chemotherapy at 12-53 and 12-55.

Response: The dichotomization of a study population according to the distribution of a determinant of interest is common for exploratory studies like ours, especially here since no prior information on optimal timing of nivolumab preexisted. Therefore 12:54 was a statistically defined cut-off time. Our study clearly showed that nivolumab was more effective in the morning group, yet the determination of the optimal circadian time window for nivolumab administration represents the next step of our research.

  1. Given the significant difference in survival rates, suspiciously few scientists have investigated this issue in the literature. What is the reason for this in your opinion?

Response: In general, the relevance of dosing time for cancer treatments has been  underestimated in oncology, with temporal logistics organization of hospitals predominating over temporal changes in biological functions. Education of the medical community to circadian clocks and their relevance for health and treatment effects has been largely lacking. Finally, the drug companies that have developed immune checkpoint inhibitors have not investigated whether timing could impact efficacy (or have not reported their results on this issue). The lack of mandatory investigations on drug timing before drug registration (or in postmarketing studies) is also a critical consideration that let people believe that immunotherapy timing would have no real effect. Our results, and those of Qian et al [26] clearly highlight that this bottleneck can and need be further circumvented through the undertaking of randomized clinical trials. We have added this paragraph in the discussion (Lines 427 – 438).

Round 2

Reviewer 1 Report

The revised paper highlights better the limits of the study.

Reviewer 3 Report

The authors did a great job and overcome many obstacles. I do appreciate this.

This manuscript is a resubmission of an earlier submission. The following is a list of the peer review reports and author responses from that submission.

Round 1

Reviewer 1 Report

The authors reported a retrospective mono-institutional experience showing a possible time-dependent efficacy of nivolumab in NSCLC patients.

Even if intriguing it is necessary to consider these results with great caution for many reasons:

-the long half-life of nivolumab (about 25 days) suggests a limited effect of circadian clock on its activity

-it’s a mono-institutional non randomized study

-the study did not provide for any specific laboratory analyses concerning, for example, cytokine concentrations or circadian hormones

-the lack of similar results in literature

Reviewer 2 Report

In this analysis, the authors assess the association of timing (chronomodulation) of nivolumab application: "Morning" vs. "Afternoon" with survival outcome parameters and demonstrate markedly superior outcome in the "Morning" group. I commend the authors for tackling chronomodulated application with ICIs, a topic they’ve also previously addressed in other anticancer drugs. There are however some major issues with this analysis and especially in the case of ICIs:

  1. The dichotomization into the "Morning" and "Afternoon" group was done based on the median time of application (timing of >50% of infusions) and is somewhat flawed as although the coefficient of variation was also included, there is still a potential for significant timing overlap between the 2 groups, Thus the use of median dichotomization does not yield homogeneous groups. However, I understand that a more homogeneous grouping is a difficult task in this setting.
  2. The groups were (borderline) significantly unbalanced regarding Age, Sex, PS and PD-L1 status. It is well-known that efficacy of ICIs is dependent on these factors and the grouping was unbalanced in all cases to the detriment of the "Afternoon" group.
  3. Another major issue was the median number of treatment courses was highly significantly different between the 2 groups: 19 (3-133) vs. 6 (3-30) in the "Morning" and "Afternoon" groups, respectively. The authors attempted to tackle this issue by presenting a 2-, 4- and 6-month landmark analysis. Where they actually show a consistent PFS/OS benefit in the "Morning" group. However, I assume the sample size in the "Afternoon" arm of the 2-, 4- and 6-month landmark analysis is too small to draw any meaningful conclusions. Please include numbers at risk for the landmark analyses (Figure 3) and p-values in each case
  4. To conduct the multivariate analysis, the patient number is quite small as evidenced by the wide 95%Cis in the case of the "Prognostic factors of OS". The rule of thumb is that Cox models should be used with a minimum of 10 events per predictor variable. Perhaps the authors should only include the UVA in this case.
  5. In contrast to other anticancer drugs, as the authors duly note, understanding the main mechanisms of nivolumab chronotherapy is challenging as a priori pharmacokinetics would suggest a limited circadian time dependency. In summary, while this is a thorough analysis, I think the findings are mainly due to the issues raised above but would implore the authors to continue their active research in this topic and perhaps attempt to dichotomize patients into more homogeneous groups even perhaps within the ramifications of a small prospective trial.

Minor:

Page 1 – Simple Summary – line 30: edit "NSCC" to "NSCLC"

Page 3 – line 106 – Include the IRB approval number

Reviewer 3 Report

Very nice article about a very challenging subject. How did the study/project started? Did you have a clear goal and an idea ? how did you decide the sample size? and what weaknesses, except for the retrospective analyses can you see? Who made the decision of choosing morning or afternoon procedure? or was it a analyses of your routines and the patients were divided into two gruops afterwards retrospectively?

Are there any animal data supporting this hypothesis? where there any bloodsamples for possible later analyses collected from the study participants?

Reviewer 4 Report

The authors presented the manuscript Time-dependent efficacy of checkpoint inhibitor nivolumab in metastatic non-small cell lung cancer patients for peer review. They state that circadian timing could play a decisive role in nivolumab efficacy, as previously shown for toxicity and/or efficacy of chronomodulated chemotherapy in colorectal and lung cancer patients Improving the efficacy of current therapeutics is of great concern. Previous randomised controlled trials have demonstrated a large and statistically significant prolongation of both progression-free survival (PFS) and overall survival (OS) with antibodies directed against PD-1 or PD-L1 as compared to cytotoxic chemotherapy in lung cancer patients. Most important finding is that morning nivolumab nearly quadrupled both median PFS and median OS as compared to afternoon dosing for patients in each tumor PD-L1 category. Manuscript is very interesting and important for clinicians.

I hve several suggestions to you.

First, have you access body temperature time course during first 24 h after drug implementation?

Second, have you provide any circadian markers to prove timing efficacy (morning vs evening)?

THird, how could you exaple lack of randomization in your study?

Fourth, have the sex of patients influenced the efficacy of nivolumab?

Reviewer 5 Report

1. I did not see information about the presence of driver mutations (in particular, EGFR, ALK). 2. A two-component chemotherapy regimen with the inclusion of platinum preparations has been the basis for the systemic treatment of patients with advanced NSCLC for many years who have not found driver mutations. However, in individuals with high levels of PD-L1 expression, immunotherapy may be more effective than chemotherapy. Why do all patients receive the same treatment regardless of their PD-L1 expression level? 3. Do survival rates depend on first-line chemotherapy? from the localization of metastases? 4. Have patients with bone metastases received bisphosphonates? 5. The obtained results in connection with the small sample and heterogeneous groups can be considered only as preliminary, therefore I recommend the authors to make the appropriate changes in the title of the article.